# Using microbiological data to improve the use of antibiotics for respiratory tract infections: A protocol for an individual patient data meta-analysis

Irene Boateng[1], Beth Stuart[1,2]*, Taeko Becque[1], Bruce Barrett[3], Jennifer Bostock[1], Robin Bruyndonckx[4], Lucy Carr-Knox[2], Emily J. Ciccone[5], Samuel Coenen[6,7], Mark Ebell[8], David Gillespie[9], Gail Hayward[10], Katarina Hedin[11,12], Kerenza Hood[9], Tin Man Mandy Lau[9], Paul Little[1], Dan Merenstein[13], Edgar Mulogo[14], Jose Ordóñez-Mena[10], Peter Muir[15], Kirsty Samuel[1], Nader Shaikh[16], Sharon Tonner[10], Alike W. van der Velden[17], Theo Verheij[17], Kay Wang[1], Alastair D. Hay[18‡], Nick Francis[1‡]

1 Primary Care Research Centre, Faculty of Medicine, University of Southampton, Southampton, United Kingdom, 2 Centre for Evaluation and Methods, Wolfson Institute of Population Health, Queen Mary University of London, London, United Kingdom, 3 Dept of Family Medicine, University of Wisconsin, Madison, WI, United States of America, 4 Data Science Institute, I-BioStat, Hasselt University, Martelarenlaan, Hasselt, Belgium, 5 Division of Infectious Diseases, University of North Carolina School of Medicine, Chapel Hill, North Carolina, United States of America, 6 Centre for General Practice, Department of Family Medicine and Population Health, University of Antwerp, Antwerp, Belgium, 7 Laboratory of Medical Microbiology, Vaccine & Infectious Disease Institute, University of Antwerp, Antwerp, Belgium, 8 Department of Epidemiology and Biostatistics, College of Public Health, University of Georgia, Athens, Georgia, United States of America, 9 Centre for Trials Research, School of Medicine, Cardiff University, Cardiff, Wales, United Kingdom, 10 Nuffield Department of Primary Care, University of Oxford, Oxford, United Kingdom, 11 Futurum, Region Jönköping County, Sweden, 12 Department of Health, Medicine and Caring Sciences, Linköping University, Linköping, Sweden, 13 Dept of Family Medicine, Georgetown University, Washington DC, United States of America, 14 Department of Community Health, Faculty of Medicine, Mbarara University of Science and Technology, Mbarara, Uganda, 15 UK Health Security Agency South West Regional Laboratory, Southmead Hospital, Bristol, United Kingdom, 16 School of Medicine, University of Pittsburgh, Pittsburgh, Pennsylvania, United States of America, 17 Julius Center for Health Sciences and Primary Care, University Medical Center Utrecht, Utrecht, the Netherlands, 18 Centre for Academic Primary Care, Bristol Medical School: Population Health Sciences, University of Bristol, Bristol, United Kingdom

‡ ADH and NF are joint senior authors on this work.
* b.l.stuart@qmul.ac.uk

## Abstract

### Background

Resistance to antibiotics is rising and threatens future antibiotic effectiveness. 'Antibiotic targeting' ensures patients who may benefit from antibiotics receive them, while being safely withheld from those who may not. Point-of-care tests may assist with antibiotic targeting by allowing primary care clinicians to establish if symptomatic patients have a viral, bacterial, combined, or no infection. However, because organisms can be harmlessly carried, it is important to know if the presence of the virus/bacteria is related to the illness for which the patient is being assessed. One way to do this is to look for associations with more severe/prolonged symptoms and test results. Previous research to answer this question for acute

**Data Availability Statement:** This is a protocol so no datasets were generated or analysed during the current study.

**Funding:** This study/project is funded by the National Institute for Health and Care Research (NIHR) School for Primary Care Research (project reference 589). The views expressed are those of the author(s) and not necessarily those of the NIHR or the Department of Health and Social Care. The funders did not and will not have a role in study design, data collection and analysis, decision to publish, or preparation of the manuscript.

**Competing interests:** The authors have declared that no competing interests exist.

respiratory tract infections has given conflicting results with studies has not having enough participants to provide statistical confidence.

## Aim

To undertake a synthesis of IPD from both randomised controlled trials (RCTs) and observational cohort studies of respiratory tract infections (RTI) in order to investigate the prognostic value of microbiological data in addition to, or instead of, clinical symptoms and signs.

## Methods

A systematic search of Cochrane Central Register of Controlled Trials, Ovid Medline and Ovid Embase will be carried out for studies of acute respiratory infection in primary care settings. The outcomes of interest are duration of disease, severity of disease, repeated consultation with new/worsening illness and complications requiring hospitalisation. Authors of eligible studies will be contacted to provide anonymised individual participant data. The data will be harmonised and aggregated. Multilevel regression analysis will be conducted to determine key outcome measures for different potential pathogens and whether these offer any additional information on prognosis beyond clinical symptoms and signs.

## Trial registration

PROSPERO Registration number: CRD42023376769.

## Introduction

Antimicrobial resistance is a global challenge [1, 2]. In the UK in 2019, 85% of antibiotics were prescribed in primary care [2], most commonly for respiratory infections, despite studies showing they are largely ineffective [3–6]. It is possible that there may be some patients with respiratory infections who may benefit [7], either by reducing the duration and severity of symptoms, and/or by preventing complications. A key challenge for primary care clinicians is identifying these patients [8].

In principle, two approaches can be used to identify the patients most likely to benefit from antibiotics [9]. The first is to see if specific patient characteristics can be identified which predict infection prognosis or etiology, and then target treatment towards these predictors [10–12]. The second is to also investigate the role of microbiology, and in particular the role of viruses and bacteria. Microbiology is a discipline largely developed in secondary care where infections are more severe and invasive samples can be which are ethically and practically unjustifiable in primary care where infections are more likely to be self-limiting. Additionally, standard laboratory microbiology takes too long to inform primary care clinical decision making [13].

Although there is much ongoing work in this area, there is currently no affordable and easily adoptable method to distinguish between a viral and bacterial respiratory infection and hence determine who may benefit from antibiotics, nor to identify the causative pathogen(s) and antimicrobial sensitivity profile. Furthermore, even when samples are collected, it can be difficult to determine whether an isolated organism is commensal or pathogenic [14].

Development of rapid microbiological point-of-care tests is now accelerating, spurred further by the SARS-CoV-2 pandemic. Some projections suggest the industry could value US

$50billion by 2025 [15]. However, whilst testing may become feasible, there remains considerable uncertainty around the clinical significance either for advising patients about prognosis or for determining the need for treatments, including antibiotics.

It is therefore critical that we develop the evidence base for the role of identifying the type of respiratory tract infections in the primary care setting, with a particular emphasis on their prognostic significance. This understanding could help guide the future use of point-of-care tests as antimicrobial stewardship tools. Ultimately, this could fundamentally change how clinicians approach antibiotic prescribing for common infections and the interaction between patients and clinicians in these consultations.

A recent systematic review, which included both primary care and secondary care settings, suggested a potential relationship between duration of hospitalisation and microbiology, with longer duration when respiratory syncytial virus (RSV), adenovirus or influenza were detected [16]. However there were very high levels of between-study heterogeneity which limited analyses. And whilst data had been collected on a large number of possible associations between microbiology and outcome, very few of these were formally evaluated or reported [16].

There have been several secondary analyses of clinical trials and observational studies in a primary care setting that have explored the relationship between microbiological data from the baseline consultation and clinical outcomes. The Cough Complication Cohort (3C) cohort study of adults presenting with acute cough suggested that those patients with a virus or bacteria detected on throat swab had poorer prognosis than those with no pathogen detected but power was limited to understand the implications of those with co-isolation of a virus and a bacterium [17]. Data from The Genomics to Combat Resistance against Antibiotics in Community-acquired LRTI in Europe (GRACE) prospective cohort and clinical trial of adults presenting with cough also suggest a longer duration of illness in those patients where a virus was detected [18]. and treatment seemed most likely to prevent deterioration when a combination of both viral and bacterial pathogens had been detected [19]. But again, it was not possible to compare between the viral, bacterial and co-isolation groups due to small numbers. The ARTIC-PC trial of children presenting with cough suggested no relationship between microbiology and prognosis [20]. Similarly, the ALiCE trial data from participants with flu-like illness showed no difference in disease course or complications between viral and bacterial infections [21]. Finally, the TARGET study of children presenting to primary care with acute cough found children in whom at least one virus was isolated had more severe symptoms on days 2 to 4 than those without any virus, and that those in whom RSV, Influenza A or Influenza B had the most severe symptoms [22].

These data suggest that the presence of virus and/or a combination of a virus and bacteria may have an effect on the patient's prognosis and that knowing this information at the time of the consultation might assist management, especially around safety netting and appropriate prescribing or self-management. But not all studies have shown results in the same direction. These studies were originally powered for other outcomes and not all participants may have provided samples. The authors themselves acknowledge that these studies are likely to be underpowered to detect any true relationships between infection type and outcomes or interactions with antibiotic treatment.

The anonymised individual patient data (IPD) meta-analysis we are proposing would help to address these gaps. By systematically reviewing the literature and then bringing together all the eligible data, we would substantially increase the power for the comparisons of interest. Having access to the individual patient data will also allow us to reduce heterogeneity by calculating outcomes and exposures in a consistent manner.

## Methods

### Aims

To undertake a synthesis of IPD from both randomised controlled trials (RCTs) and observational cohort studies of acute respiratory tract infections (RTI) in order to investigate the prognostic value of microbiological data in addition to, or instead of, clinical symptoms and signs.

### Objectives

- To use individual patient data from both RCTs and observational studies to determine key outcome measures (duration of illness, severity of illness, probability of developing complications/requiring repeated consultations) for different infection types (viral, bacterial and combined infection) detected in microbiological samples from patients with acute RTI, and whether these offer any additional prognostic information beyond symptoms, signs and examination findings routinely collected in the consultation.

- To explore whether there are interactions between antibiotic treatment and infection type with respect to these key clinical outcomes. This will potentially enable us to determine whether there are subgroups of patients who can be identified by their baseline microbiological data as not being suitable candidates for antibiotic prescribing and therefore assisting primary care clinicians in making more appropriate prescribing decisions for individual patients presenting with an RTI.

- To explore whether there are clusters or combinations of individual organisms and to explore whether these are predictive of prognosis.

### Study approach for the systematic review to identify studies

A full systematic search of the literature will be conducted to identify relevant randomized controlled trials and observational cohort studies that might be able to contribute data based on the following criteria:

**Population.**   All patients attending a primary or community care setting with an acute RTI, including urgent care and emergency departments.

**Prognostic factors.**   Microbiological data on infection type, classified as none, virus, bacterium and combined virus/bacterium. We will also explore the role of individual organisms, where the data allows. We will explore whether these have prognostic value above symptoms, signs and examination findings during baseline consultation.

**Outcomes.**   Duration of illness, severity of illness, repeated consultation with new or worsening illness, complications requiring hospitalisation or leading to death.

Studies of secondary care inpatients will be excluded as will studies which are not RCT or observational cohorts (e.g. cross-sectional, case control or survey studies).

### Search strategy for identification of studies

We will base the search criteria on two previous systematic reviews carried out by our co-authors [16, 23]. The search terms have been updated to include observational cohort studies and to include both adults and children. We will also contact content experts to determine whether there are unpublished studies of either type that might be able to provide data. Full search criteria are set out in the S1 Appendix.

We will search Medline, Embase and Cochrane library from January 2008 to November 2022. We have limited the search to the last 15 years consistent with IPD data availability and

to ensure that the included patient population reflects those currently consulting in primary care.

Search results will be exported to and deduplicated in Endnote before being uploaded to Rayyan for screening. One team member (IB) will screen all titles and abstracts, with at least 20% screened by a second reviewer (BS). Full text articles will be uploaded to Rayyan and two reviewers (IB and BS) will independently assess their eligibility. All discrepancies will be resolved by discussion with a third reviewer (NF or AH).

## Risk of bias assessment and certainty of evidence assessment

We will use the Quality In Prognostic Studies (QUIPS) tool [24] to assess the quality of included studies. The aspects of the studies which will be assessed are:

1. The participants

2. Study attrition

3. Prognostic factor measurement

4. Outcome measurement

5. Study confounding

6. Statistical analysis and reporting

Given that the studies to be included in the review will not primarily have been designed to explore questions around the prognostic value of microbiological data it may be difficult to evaluate some domains. Therefore, in addition to the 4-grade scale (yes, partial, no, unsure) we will add a "not applicable" grading. Not applicable items will not count towards overall risk of bias assessment for that domain. If a whole domain cannot be assessed, it will be recorded as "not applicable".

Two reviewers will independently assess the risk of bias and resolve disagreement by discussion.

We will use GRADE to rate the overall certainty (quality) of evidence that includes the evaluation of risk of bias, inconsistency, indirectness, imprecision and publication factors.

## Data extraction and database creation

Authors of eligible studies will be contacted and asked whether they would be willing to collaborate in this study and share their data. Data will be considered unavailable if none of the authors have responded after three attempts or if the authors indicate that they are unable to share data.

Authors will be asked to provide data in a format acceptable to them. Wherever possible, collaborators will be asked to provide the complete dataset rather than key variables used in publication. This will potentially allow us to more fully explore the microbiological data with respect to characteristics such as microbial density and volume rather than simply the presence of absence of particular organisms, the impact of confounding variables and any interactions with antibiotic treatment. It will also allow us to harmonise the covariates and outcomes to ensure that a consistent approach has been used across studies. This will in turn reduce the level of between-study heterogeneity. We will also request that authors share details of the approach to assessing microbiological outcomes or provide a lab manual or protocol if available.

Data will be checked by comparing key variables with published data and collaborators contacted if important discrepancies are identified and asked to clarify. The level of missing data

will be assessed and discussed with study collaborators. If less than 5% of data is found to be missing on baseline characteristics and outcome measures, the data will be analysed on a complete cases basis. However, if there is substantial missing data, the pattern and nature of the missingness will be explored and multiple imputation by chained equations (MICE) will be used if appropriate. We will also explore the pattern and nature of missing microbiological data and will consider whether multiple imputation may be an appropriate strategy. A copy of the data to be included in the IPD analyses will be converted to Stata (version 15.0 or higher) for analysis. This dataset will include: study id, participant id, country, outcome measures, microbiological data, sociodemographic data, comorbid conditions, and signs, symptoms and examination findings collected at baseline consultation.

## Analysis

We will describe study and patient characteristics for all studies that contribute IPD, as well as those that declined or were unable to provide data. This will help to determine the extent to which the included studies are a representative sample of patients presenting in primary care with acute RTI.

IPD meta-analysis can be conducted using either a two-stage or a one-stage approach. The two-stage approach involves calculating the effect for each study separately and then combining the results using traditional meta-analysis techniques. The one-stage approach combines all the data in a single meta-analysis based on a regression model stratified by study. One stage analysis is often more appropriate for exploring treatment-covariate interactions as it has increased power and is less likely to suffer from aggregation bias [25]. Given that one of the key aims of this study is to explore interactions between antibiotic treatment and pathogens, we will first use the one-step approach.

All analyses will be repeated using a two-step approach as a sensitivity analysis in order to allow us to include and explore the impact of the aggregate results from those studies where we are unable to obtain data, either because the authors declined to collaborate or because the IPD was no longer available, assuming the relationship between microbiological data and outcomes was reported in the published paper.

Heterogeneity will be summarised using the $I^2$ statistic. Sources of heterogeneity will be explored and explained as full as possible and a random effects model used for all analyses.

In the IPD dataset, we will calculate the key outcome measures on a consistent basis. Decisions regarding how to harmonise key variables will be fully reported. A regression model will be used to explore the relationship between the outcome measures and the microbiological data as a categorical variable (viral, bacterial, combined infection) with "no organism detected" as the reference category. The model selected will be one that is appropriate to the outcome measure of interest. These are most likely to be linear regression to model the severity of symptoms following consultation, a negative binomial model to assess the duration of illness and logistic regression for the repeat consultation and complications outcomes. All models will control for age, baseline symptom severity, symptom duration prior to consultation, country and antibiotic strategy. All models will allow for the clustering of participants within studies by including a random effect for study. We will present overall pooled results for all RTIs as well as for subgroups of diagnoses: sore throat, sinusitis, otitis media and LRTI.

Both RCTs and observational studies are potentially subject to confounding in this context. Whilst interventions may be randomised in the RCT context, the microbes themselves are not and not all RCTs will have randomised the treatment as some antimicrobial stewardship interventions may focus on randomising the delivery of care (e.g. CRP testing) rather than antibiotic treatment. And whilst observational studies may be more inclusive of the patient

population, they are often at higher risk of bias. We will therefore carefully consider for both study types how best to control for potential confounders, including the use of propensity score analysis, a technique developed to help balance studies on key measured covariates. Whilst the primary analysis will therefore be of the pooled RCT and observational study data-sets after controlling for potential confounders, due to their different designs, the results of RCT and observational studies will be reported separately as a sensitivity analysis. If there is sufficient data, we may conduct a sensitivity analysis which treats those RCTs which randomized the delivery of care as a separate group to those which randomized the antibiotic treatment.

## Subgroups and additional analyses

We will explore whether microbiology adds any additional prognostic value to baseline symptoms, signs and examination findings, treating the models as nested and using a likelihood ratio test for all outcomes. Exploratory analyses will, where data allows, repeat these analyses for individual microbes.

We will explore whether there is an interaction with age, with particular interest in whether the inferences differ in children compared to adults, and whether there is an interaction with setting (primary care compared to urgent/emergency care).

To explore whether the relationship between infection type and outcome is modified by antibiotic prescribing, the analyses described above will be repeated, testing for interactions between antibiotic treatment (immediate/delayed/none) and pathogen (viral/bacterial/combined/none). If there is sufficient data, we will also undertake a sensitivity analysis using data only from those individuals in clinical trials who were not prescribed antibiotics.

We will also request data on microbial density, such as colony counts or PCR cycle time. If sufficient data is available and can be harmonised, analyses will be repeated using this continuous measure and controlling for the method of collection used.

We will describe the most common co-infection profiles. If there is sufficient data, we will undertake latent class analysis to explore whether it is possible to detect clusters of patients who are similar with respect to the micro-organisms detected (present/absent) during their illness. We can then explore whether these classes are predictive of the outcomes set out above. This may help to determine which organisms are commensal and which are pathogenic.

Latent class models will be fitted successively starting with a one-cluster model and then adding another cluster for each successive model. The best-fitting model will be selected based on 2 criteria: (1) the Bayesian Information Criterion (BIC) (where a lower BIC indicates better fit) and (2) the average posterior probabilities of group membership, as a measure of classification quality ($>0.80$ or greater in all groups) [26].

## Public and patient involvement

The study has two patient and public involvement (PPI) co-applicants, JB and KS, both of whom have experienced recurrent infections and have been inappropriately prescribed antibiotics. They have been involved in shaping the research questions and will continue to contribute to all stages of the research. Both have reviewed and written sections of the funding application and are coauthors on this manuscript. As the study progresses, they will be involved in discussion of data harmonisation and the interpretation of study findings. In return for their time, PPI team members will be reimbursed in line with National Institute of Health Research (NIHR) recommendations and will be co-authors on research outputs where appropriate. They will also be integral to the dissemination of our findings to the wider public.

We will also convene a diverse PPI panel of 10+ people with a history of acute RTI as patients and/or carers to discuss the results as they emerge. This will help us to determine how best to interpret and disseminate the findings in a way that is meaningful to patients and the public and to consider the priorities for future research.

## Dissemination

We will publish the results of this study in high impact, open access, peer reviewed academic journals, and submit abstracts to present the results at national and international conferences, especially those aimed at primary care practitioners. Along with our PPI team members, we will explore how best to share our results more widely to the public and patient populations. We plan on interacting with the NIHR Community Healthcare MIC's network of diagnostics industry collaborators and use our links with the Longitude Prize to ensure that the findings feed directly into diagnostics development, ensuring that the correct pathogens are prioritised for primary care focused novel diagnostic development.

## Supporting information

**S1 Checklist. PRISMA-P 2015 checklist.**
(DOCX)

**S1 Appendix. Search terms.**
(DOCX)

## Author Contributions

**Conceptualization:** Beth Stuart, Jennifer Bostock, Gail Hayward, Kerenza Hood, Tin Man Mandy Lau, Paul Little, Peter Muir, Kirsty Samuel, Alastair D. Hay, Nick Francis.

**Data curation:** Bruce Barrett, Robin Bruyndonckx, Lucy Carr-Knox, Emily J. Ciccone, Samuel Coenen, Mark Ebell, David Gillespie, Gail Hayward, Katarina Hedin, Dan Merenstein, Edgar Mulogo, Jose Ordóñez-Mena, Nader Shaikh, Sharon Tonner, Alike W. van der Velden, Theo Verheij, Kay Wang.

**Funding acquisition:** Beth Stuart, Jennifer Bostock, David Gillespie, Gail Hayward, Kerenza Hood, Tin Man Mandy Lau, Paul Little, Peter Muir, Kirsty Samuel, Alastair D. Hay, Nick Francis.

**Methodology:** Beth Stuart, Taeko Becque, Robin Bruyndonckx, Lucy Carr-Knox, Samuel Coenen, David Gillespie, Gail Hayward, Kerenza Hood, Tin Man Mandy Lau, Peter Muir, Alastair D. Hay, Nick Francis.

**Project administration:** Irene Boateng.

**Resources:** Irene Boateng, Beth Stuart.

**Supervision:** Beth Stuart, Taeko Becque, Nick Francis.

**Writing – original draft:** Irene Boateng, Beth Stuart, Alastair D. Hay.

**Writing – review & editing:** Irene Boateng, Beth Stuart, Taeko Becque, Bruce Barrett, Jennifer Bostock, Robin Bruyndonckx, Lucy Carr-Knox, Emily J. Ciccone, Samuel Coenen, Mark Ebell, David Gillespie, Gail Hayward, Katarina Hedin, Kerenza Hood, Tin Man Mandy Lau, Paul Little, Dan Merenstein, Edgar Mulogo, Jose Ordóñez-Mena, Peter Muir, Kirsty Samuel, Nader Shaikh, Sharon Tonner, Alike W. van der Velden, Theo Verheij, Kay Wang, Alastair D. Hay, Nick Francis.

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
