## [Decision Letter · Decision Letter 0]

25 Sep 2023

PONE-D-23-21834Using microbiological data to improve the use of antibiotics for respiratory tract infections: a protocol for an individual patient data meta-analysisPLOS ONE

Dear Dr. Stuart,

Thank you for submitting your manuscript to PLOS ONE. After careful consideration, we feel that it has merit but does not fully meet PLOS ONE’s publication criteria as it currently stands. Therefore, we invite you to submit a revised version of the manuscript that addresses the points raised during the review process.

We look forward to receiving your revised manuscript.

Kind regards,

Fran

Francis John Gilchrist

Academic Editor

PLOS ONE

Additional Editor Comments:

Please carefully address the comments made by the two reviewers.

Reviewers' comments:

Reviewer's Responses to Questions

**Comments to the Author**

1. Does the manuscript provide a valid rationale for the proposed study, with clearly identified and justified research questions?

Reviewer #1: Yes

Reviewer #2: Yes

2. Is the protocol technically sound and planned in a manner that will lead to a meaningful outcome and allow testing the stated hypotheses?

Reviewer #1: Yes

Reviewer #2: Yes

3. Is the methodology feasible and described in sufficient detail to allow the work to be replicable?

Reviewer #1: Yes

Reviewer #2: Yes

4. Have the authors described where all data underlying the findings will be made available when the study is complete?

Reviewer #1: No

Reviewer #2: Yes

5. Is the manuscript presented in an intelligible fashion and written in standard English?

Reviewer #1: Yes

Reviewer #2: Yes

6. Review Comments to the Author

You may also provide optional suggestions and comments to authors that they might find helpful in planning their study.

Reviewer #1: Thank you for the invitation to review this protocol manuscript. My only two comments/ suggestions are

- The authors have not described where all data underlying the findings will be made available when the study is complete as stipulated by the journal

- Have the authors considered analysis of studies conducted in urgent care and emergency departments separately as patients who attend these settings may have features in common with secondary care, such as the severity of illness or may be distinct from primary care populations, for example there may be larger proportions of homeless/ unregistered patients who may not be able to access primary care clinics

Reviewer #2: Thank you for the opportunity to review the manuscript. This manuscript addresses a very important topic that will contribute greatly to the management of common acute infections and to the global primary care antimicrobial stewardship efforts.

I have a few minor comments for authors to consider improving the protocol.

• Search strategy

• The authors have mentioned in Page 12, Line 32 that they will search databases from 2008-2022. It is unclear why authors are limiting their search to studies from 2008? If it is because authors have existing data before 2008, I recommend clarifying.

• Methods

o Page 12, Line 11: Population: The authors mention that the study will include patients attending primary or community care setting with RTI including urgent care and emergency setting. Would they consider those at outpatients’ hospitals? I suggest clarifying to help capture possible studies in LMICs that targeted at hospital outpatients.

o Inclusion criteria: I suggest clarifying studies that will be included? Will authors be assessing RCTs comparing antibiotics with placebo or no treatment, other treatments with placebo?

• Result:

o Page 14, Line 41: Analysis of the result. Will the analysis include all RTI as a cluster or subgroup analysis will highlight outcome of specific conditions like otitis media, sore throat?

o References: Kindly check reference 1, 25 to ensure they are correctly cited

7. PLOS authors have the option to publish the peer review history of their article (what does this mean?). If published, this will include your full peer review and any attached files.

Reviewer #1: **Yes: **Uy Hoang

Reviewer #2: **Yes: **Kwame Peprah Boaitey

---

## [Author Response · Author response to Decision Letter 0]

16 Oct 2023

We would like to thank the reviewers for their time in reviewing our protocol and their helpful comments. We have responded to the specific reviewer comments below, indicating where amendments have been made to the manuscript. Please note that in formatting the manuscript in line with the journal guidelines, the page and line numbering will have changed. 

Reviewer #1: Thank you for the invitation to review this protocol manuscript. My only two comments/ suggestions are

- The authors have not described where all data underlying the findings will be made available when the study is complete as stipulated by the journal

At this stage, this is only a protocol paper so there are no findings and hence there is no data relevant to this PLOS One publication to share. Once the study is complete, the IP for the data underlying the IPD analysis still remains with the original authors as per the legal agreements that allow the data to be shared. It is therefore not possible to make the individual level dataset available in a repository. We will explore prior to publication of the study results whether an aggregate level dataset and any analysis code could be shared.

- Have the authors considered analysis of studies conducted in urgent care and emergency departments separately as patients who attend these settings may have features in common with secondary care, such as the severity of illness or may be distinct from primary care populations, for example there may be larger proportions of homeless/ unregistered patients who may not be able to access primary care clinics

Thank you for this suggestion. We have added this as a sensitivity analysis if there is sufficient data. The text on page 14, lines 3-5 now read: We will explore whether there is an interaction with age, with particular interest in whether the inferences differ in children compared to adults, and whether there is an interaction with setting (primary care compared to urgent/emergency care)

Reviewer #2: Thank you for the opportunity to review the manuscript. This manuscript addresses a very important topic that will contribute greatly to the management of common acute infections and to the global primary care antimicrobial stewardship efforts.

I have a few minor comments for authors to consider improving the protocol.

• Search strategy

• The authors have mentioned in Page 12, Line 32 that they will search databases from 2008-2022. It is unclear why authors are limiting their search to studies from 2008? If it is because authors have existing data before 2008, I recommend clarifying.

There were two main reasons for this choice. Firstly, we have seen in changes in consulting behaviour leading to many patients attending earlier in their illnesses. Secondly, IPD meta-analysis relies on data availability. Whilst data may be retained in line with institutional policies for 10-15 years, beyond this point it is usually destroyed or lost. We felt it would take a lot of effort to trace older datasets and their owners and that was unlikely that this additional effort would lead to our being able to obtain individual level data. We therefore restricted our search for data to the last 15 years. 

We have added the following to the text on page 9 (Search strategy subheading): We have limited the search to the last 15 years consistent with IPD data availability and to ensure that the included patient population reflects those currently consulting in primary care. 

• Methods

o Page 12, Line 11: Population: The authors mention that the study will include patients attending primary or community care setting with RTI including urgent care and emergency setting. Would they consider those at outpatients’ hospitals? I suggest clarifying to help capture possible studies in LMICs that targeted at hospital outpatients.

We have included outpatient hospitals if the participants have consulted for RTI in an acute capacity. We have not included inpatients or routine outpatient clinics. This has identified a number of possible studies in LMICs so we believe the search terms are picking up the relevant papers. 

o Inclusion criteria: I suggest clarifying studies that will be included? Will authors be assessing RCTs comparing antibiotics with placebo or no treatment, other treatments with placebo?

We have included RCTs in primary care if patients consulted for acute RTI and had a microbiological sample taken at baseline consultation, regardless of the randomised comparison. The intervention in the original study does not form part of the PICO for inclusion in this IPD. So some will include comparisons of treatments with placebo, with one another, or other types of stewardship interventions. 

 • Result:

o Page 14, Line 41: Analysis of the result. Will the analysis include all RTI as a cluster or subgroup analysis will highlight outcome of specific conditions like otitis media, sore throat?

This is an important consideration and we appreciate the reviewer raising it. We have discussed and have added some clarification to our analysis. We will include all RTI to achieve a pooled effect, as the prevailing diagnosis does not preclude other symptoms from being present – otitis media and sore throat can occur in the same patient – but we will present subgroup analyses by main diagnosis. This should allow us to highlight outcomes for specific conditions and explore any heterogeneity by diagnosis. 

We have added the following to the text (page 13, line 3): We will present overall pooled results for all RTIs as well as for subgroups of diagnoses clustered by whether they are patients presenting with predominantly upper respiratory tract symptoms (sore throat, sinusitis, otitis media) or “other” to include cough, ILI and other lower respiratory tract infections.

o References: Kindly check reference 1, 25 to ensure they are correctly cited

Thank you – we have amended these references. The numbering of references has slightly changed as we spotted some which were not correctly linked to statements within the text.

---

## [Decision Letter · Decision Letter 1]

10 Nov 2023

Using microbiological data to improve the use of antibiotics for respiratory tract infections: a protocol for an individual patient data meta-analysis

PONE-D-23-21834R1

Dear Dr. Stuart,

We’re pleased to inform you that your manuscript has been judged scientifically suitable for publication and will be formally accepted for publication once it meets all outstanding technical requirements.

Kind regards,

Fran

Dr FJ Gilchrist

Academic Editor

PLOS ONE

Additional Editor Comments (optional):

Reviewers' comments:

Reviewer's Responses to Questions

**Comments to the Author**

1. Does the manuscript provide a valid rationale for the proposed study, with clearly identified and justified research questions?

Reviewer #1: Yes

Reviewer #2: Yes

2. Is the protocol technically sound and planned in a manner that will lead to a meaningful outcome and allow testing the stated hypotheses?

Reviewer #1: Yes

Reviewer #2: Yes

3. Is the methodology feasible and described in sufficient detail to allow the work to be replicable?

Reviewer #1: Yes

Reviewer #2: Yes

4. Have the authors described where all data underlying the findings will be made available when the study is complete?

Reviewer #1: Yes

Reviewer #2: Yes

5. Is the manuscript presented in an intelligible fashion and written in standard English?

Reviewer #1: Yes

Reviewer #2: Yes

6. Review Comments to the Author

You may also provide optional suggestions and comments to authors that they might find helpful in planning their study.

Reviewer #1: Thank you for your resubmission. I have no further comments or queries suggestions for the manuscript.

Reviewer #2: Thank you for the opportunity to re-examine the manuscript “Using microbiological data to improve the use of antibiotics for respiratory tract infections: a protocol for an individual patient data meta-analysis”.

After reviewing authors responses to my initial comments, I am pleased to see that the authors have addressed most of the concerns raised and have provided clear answers to my queries.

I believe the authors have addressed all other questions related to PLOS ONE publication guidance.

I have no new comment.

Thank you.

7. PLOS authors have the option to publish the peer review history of their article (what does this mean?). If published, this will include your full peer review and any attached files.

Reviewer #1: **Yes: **Uy Hoang

Reviewer #2: **Yes: **Kwame Peprah Boaitey

---

## [Editor Report · Acceptance letter]

14 Nov 2023

PONE-D-23-21834R1 

Using microbiological data to improve the use of antibiotics for respiratory tract infections: a protocol for an individual patient data meta-analysis 

Dear Dr. Stuart:

I'm pleased to inform you that your manuscript has been deemed suitable for publication in PLOS ONE. Congratulations! Your manuscript is now with our production department. 

Kind regards, 

on behalf of

Dr. Francis John Gilchrist 

Academic Editor

PLOS ONE